# Analysis of Three-Dimensional Cell Migration in Dopamine-Modified Poly(aspartic acid)-Based Hydrogels

**DOI:** 10.3390/gels8020065

**Published:** 2022-01-18

**Authors:** David Juriga, Eszter Eva Kalman, Krisztina Toth, Dora Barczikai, David Szöllősi, Anna Földes, Gabor Varga, Miklos Zrinyi, Angela Jedlovszky-Hajdu, Krisztina S. Nagy

**Affiliations:** 1Department of Biophysics and Radiation Biology, Semmelweis University, H-1089 Budapest, Hungary; toth.krisztina.105@gmail.com (K.T.); barczikai.dora@med.semmelweis-univ.hu (D.B.); szollosi.sote@gmail.com (D.S.); mikloszrinyi@gmail.com (M.Z.); hajdu.angela@med.semmelweis-univ.hu (A.J.-H.); 2Department of Molecular Biology, Semmelweis University, H-1083 Budapest, Hungary; kalman.eszter@med.sememlweis-univ.hu; 3Department of Oral Biology, Semmelweis University, H-1089 Budapest, Hungary; foldes.anna@dent.semmelweis-univ.hu (A.F.); varga.gabor@dent.semmelweis-univ.hu (G.V.)

**Keywords:** poly(aspartic acid) hydrogel, dopamine, three-dimensional cell migration, hydrogel scaffold, SH-SY5Y

## Abstract

Several types of promising cell-based therapies for tissue regeneration have been developing worldwide. However, for successful therapeutical application of cells in this field, appropriate scaffolds are also required. Recently, the research for suitable scaffolds has been focusing on polymer hydrogels due to their similarity to the extracellular matrix. The main limitation regarding amino acid-based hydrogels is their difficult and expensive preparation, which can be avoided by using poly(aspartamide) (PASP)-based hydrogels. PASP-based materials can be chemically modified with various bioactive molecules for the final application purpose. In this study, dopamine containing PASP-based scaffolds is investigated, since dopamine influences several cell biological processes, such as adhesion, migration, proliferation, and differentiation, according to the literature. Periodontal ligament cells (PDLCs) of neuroectodermal origin and SH-SY5Y neuroblastoma cell line were used for the in vitro experiments. The chemical structure of the polymers and hydrogels was proved by ^1^H-NMR and FTIR spectroscopy. Scanning electron microscopical (SEM) images confirmed the suitable pore size range of the hydrogels for cell migration. Cell viability assay was carried out according to a standardized protocol using the WST-1 reagent. To visualize three-dimensional cell distribution in the hydrogel matrix, two-photon microscopy was used. According to our results, dopamine containing PASP gels can facilitate vertical cell penetration from the top of the hydrogel in the depth of around 4 cell layers (~150 μm). To quantify these observations, a detailed image analysis process was developed and firstly introduced in this paper.

## 1. Introduction

Tissue engineering is a rapidly developing and promising field that applies living cells growing on complex biocompatible matrices called scaffolds, to create tissue-like structures [1,2,3]. These scaffolds can be used for medical therapies repairing the injured or diseased area of tissues [1,2,3], but they are also suitable for testing the efficacy and toxicity of various drugs [2,4]. A high number of research projects related to tissue engineering are focusing on hydrogels, due to their structural similarity to the natural environment of the living cells, i.e., the extracellular matrix (ECM). Additionally, hydrogels have favorable mechanical properties that can be customized according to the targeted tissue type [5]. The adequate behavior of the cells can be achieved by incorporating growth or differentiation factors [6], as well as other bioactive agents like proteins, peptides, or small molecules into the hydrogel matrix [1,3,7,8].

Since several components of the ECM are amino acid-based macromolecules, i.e., proteins such as collagen or elastin, amino acid-based hydrogels can mimic the ECM, especially from the chemical point of view. However, the application of amino acid-based hydrogels as scaffolds is presently limited due to their difficult and expensive preparation [9]. Currently, there are only a few types of synthetic poly(amino acids) available, such as polylysine or poly(aspartic acid) (PASP) that can be suitable for this purpose [10,11]. In our previous studies, the chemical constitution and the physical properties of PASP-based hydrogels have been comprehensively investigated [12,13,14] and successfully tailored for applications as a scaffold [8,15]. We have demonstrated that thiol groups in the PASP hydrogels facilitate cell adhesion and proliferation [15]. Since the anhydrous form of PASP, namely poly(succinimide) (PSI) is very reactive, it can be covalently modified with various biologically active molecules (such as RGD tripeptide, taurine, doxorubicin, or dopamine) [16,17]. This feature gives us several opportunities to further improve its properties for tissue engineering and other biomedical applications [11,18].

Dopamine (DA) is one of the major neurotransmitters in the central nervous system, participating in the modulation of various functions [19] as well as the acquisition of new memories [20]. In addition, recent intense research efforts inspired by the investigation of adhesive proteins of marine mussels led to the recognition that DA also plays an important role in cell adhesion [21,22] and enhances the bioadhesive properties of different coatings and hydrogels [23,24,25]. Besides these important functions, the release of DA plays a key role in stem-cell-based tooth repair during dental pulp injury [26]. According to the literature, DA show an interesting biphasic effect on neurons. Although it is cytotoxic at high concentrations, it has been proved to have trophic or cell protective effect at low concentrations [27,28,29]. The main aim of this study is to investigate of the concentration dependent effect of the covalently bound DA in the hydrogel on the cell viability, morphology, and distribution. In accordance with this aim, SHSY-5Y and PDLC were used in the in vitro experiments, since both cell types show neurogenic activity, and the presence of dopamine receptors has been published [26].

Human wisdom teeth are novel and promising sources of postnatal stem and progenitor cells, such as PDLCs for tissue engineering purposes [30,31]. PDLCs can be isolated from the root surface of wisdom teeth by a minimally invasive intervention [32] and they have differentiation capacity into multiple cell lineages, such as fibroblasts, cementoblasts, adipocytes, osteoblasts, and chondrocytes [6,30,33], as well as neuron-like cells [31]. Furthermore, PDLCs proved to exhibit potential for regeneration of cementum, bone, periodontal fibers [32], and tendon [34], and even nerves in vivo due to their neuroectodermal origin [35]. Up to now, mostly natural polymers like collagen [36], chitosan [4], alginate/hyaluronic acid [35], or synthetic, biocompatible polymers, such as poly(glycolic acid), poly(lactic acid) [37], or poly(ethylene glycol) [38] have been suggested to prepare scaffolds for PDLCs.

As the terminally differentiated mature neurons cannot be propagated in cell culture, tumor cell lines consisting of neuroblasts (i.e., primitive nerve cells representing a previous stage of the neural differentiation process) are often used in neurobiological research. The advantages of the SH-SY5Y neuroblastoma cell line are the human origin, ease of maintenance, and well-established protocols for generating a more mature neuron-like phenotype [39,40]. Previously, this cell line was cultured mainly in scaffolds composed of natural polymers like collagen [41], bacterial nanocellulose [42], gelatin [43], or alginate/gelatin scaffolds [43].

Consequently, our study aimed to fabricate PASP-based hydrogels grafted with dopamine to combine the prosperous properties of amino acid-based polymers and dopamine in tissue regeneration. Our aims involved the characterization of these hydrogels, and in vitro studies using two relevant cell types, namely PDLCs and the SH-SY5Y neuroblastoma cell line. Thus, in the present study, DA-modified thiolated PASP hydrogels are introduced as novel scaffolds for three-dimensional cell cultivation. The hydrogel scaffolds were characterized both physically and chemically, moreover the effect of dopamine concentration on the morphology, proliferation as well as two- and three-dimensional distribution of PDLCs and SH-SY5Y cells in the hydrogels have been studied in vitro. In addition, a novel method was developed and firstly introduced in this paper to quantify the vertical penetration and migration ability of the cells.

## 2. Results and Discussion

### 2.1. Characterization of PSI, PASP Based Polymers, and PASP-Based Hydrogels

Since the preparation of the hydrogels consists of several steps (modification of the PSI with DA, crosslinking of the PSI-DA, hydrolysis of the PSI-based gels, and cleavage of the disulfide bonds to thiol groups) (Section 4.1), we aimed to determine the chemical structures of the intermediate and the final products of the preparation. To characterize the chemical structure of the PSI and PASP-based polymers after DA modification and the final PASP form of the hydrogels following the formation of the thiol groups, FTIR spectrometry was used (Figure 1a,b). Since DA can be released from the main polymer during the cleavage of the peptide bond between DA [18] and the main polymer, the DA content of the hydrogel can change during hydrolysis. As the DA content of the polymer was very low (every 10th, 20th, and 40th monomer was modified, respectively), the changes in DA concentration during hydrolysis were monitored by 1H-NMR spectroscopy (Figure 1a). Since NMR measurement cannot be used easily to determine the chemical structure of the polymer matrix, PSI-DA polymers were hydrolyzed into PASP polymers (with the same method that we applied during the preparation of the hydrogels) and the chemical structure of the PASP-based polymers were determined.

On the FTIR spectra of the PSI-DA polymers (Figure 1a, left spectra), only minor differences can be recognized between the samples. The main difference is that a new peak appeared at wavenumber 1510 cm^−1^ on the spectra of the dopamine-modified polymers. This peak is related to the stretching motion of the C–N bonds in the amide group, which proves the successful modification of the PSI main polymer. The list of the other peaks can be found in the Appendix A and in our previous publication [18].

The ^1^H-NMR spectra of the different PASP-DA polymers can be seen in Figure 1a (right spectra). The peaks at 4.59 ppm and 2.81 ppm can be assigned to methine and methylene hydrogen of aspartic acid monomer units [44,45]. The peaks at 3.36 ppm are related to the –CH_2_ unit in the dopamine, while the peaks at 6.67, 6.78, and 6.83 ppm belong to the hydrogens in the catechol group of dopamine. The peaks at 0.89, 1.34, and 1.62 ppm can be assigned to the protons in N,N-dibutyl-amine. The presence of these peaks shows that secondary amines can react with the succinimide ring, as was published in our previous articles [8,18]. The peaks at 3.15 and 5.29 ppm are connected to the methylene and methine groups in the succinimide rings, which proves that the polymers contain unreacted succinimide rings despite the hydrolysis. The residual amount of succinimide rings indicates that the applied time period for hydrolysis is not enough to produce PASP completely. Therefore, longer reaction time (several days) was applied for hydrolysis during hydrogel preparation. Due to the very small amount of modification, the exact ratios between the DA and unmodified succinimide rings cannot be determined.

The FTIR spectra of the thiolated hydrogels can be seen in Figure 1b. No significant differences can be recognized between FTIR spectra of the hydrogels due to the small amount of modification. The wide peak between wavenumbers 2800 and 4200 cm^−1^ and the disappearance of the peak at 1720 cm^−1^ prove the successful hydrolysis and the poly(aspartic acid) structure of the hydrogels. The only difference between the dopamine-containing and the DA_0_ hydrogel can be seen at wavenumber 2966 cm^−1^. This peak is related to the C–H stretching in the aromatic ring of dopamine, which proves the presence of dopamine in the polymer matrices after hydrolysis. Our spectra are similar to the spectra that can be found in the literature [14,44].

The microscopic structure of the different hydrogels was investigated by SEM (Figure 1c) and the average pore size of the freeze-dried hydrogels was determined (Table 1).

The pore size of the gels was found to be between 80 and 120 µm, which is significantly higher than the diameter of PDLCs (around 30–40 µm) [32]. This pore size range can support the vertical migration of PDLCs into the polymer matrices, as was proved by Kim and co-workers [46]. According to our results, both the average pore size (Table 1) and the mass swelling degree (Table 2) depend on the dopamine content of the hydrogel. An inverse correlation can be observed between these two properties and the dopamine content (Table 1 and Table 2). The higher dopamine content in the hydrogel resulted in weaker polymer–solvent interaction, leading to smaller mass swelling [47,48]. The smaller mass swelling degree indicates a denser polymer matrix that results in a smaller average pore size after freeze-drying the hydrogels.

### 2.2. Dopamine Release during the Hydrolysis of the Gels

According to the abovementioned physical and chemical characterization of the hydrogels by NMR and FTIR, there is remaining DA on the polymer and in the hydrogel after hydrolysis. Since we already know that DA can be released from the PASP in alkaline circumstances [18], the DA release was also investigated during the gel hydrolysis in parallel with the characterization. The effect of dopamine on different cell types is significantly concentration-dependent [49,50]; therefore, it is crucial to know the exact DA concentration to understand the results of the cell experiments. The released amount of dopamine and the dopamine concentration of the hydrogels applied in the cell experiments can be found in Table 2.

According to Table 2, approximately 50% of the dopamine was released from the hydrogel samples, which is in good correlation with our previous results regarding PSI-DA polymer conjugates [18]. Due to the DA release, the concentration of the DA in the different hydrogels was significantly changed after hydrolysis. Although the applied amount of the dopamine at the gel synthesis was 2 and 4 times higher in DA_1/10_ compared to DA_1/20_ and DA_1/40_, respectively, the DA concentration ratio of DA_1/10_ to DA_1/20_ and DA_1/40_ after the hydrolysis became 2.5 and 5, respectively. This discrepancy is a result of the different swelling degrees of the hydrogels, which can be calculated as the ratio of the swelled and dried hydrogels. The lower mass swelling degree indicates higher polymer, and therefore higher DA, content in the same amount of swelled hydrogels. Although these DA concentrations are higher than the lethal concentration for different cell types including the SH-SY5Y cells, in our case the covalent bonds between DA and the polymer matrices prolonged the release of the DA. Consequently, the free dopamine concentration in the cell culture medium was significantly lower than the calculated DA content in the hydrogels.

### 2.3. Cell Viability and Morphology on Different PASP-DA Hydrogels

Dopamine has a variety of effects on various organs: it has an essential role in normal brain function (involved in modulation of learning, memory, and coordination of movement) [19,20], and it is also an important regulator of blood pressure [51]. Dopamine exerts its effect on the cells through its cell surface receptors, which have five types: D1R, D2R, D3R, D4R, and D5R [51]. DA can enhance the adhesion of different cell types [23,24,25,52] onto various surfaces, but the effect of dopamine in vitro is highly concentration-dependent [27,53]. Low concentrations of dopamine have a cell-protective, namely anti-apoptotic effect, but at higher concentrations, it can be toxic due to the induction of apoptosis [28,49,50].

Since the main aim of this study was to investigate the concentration-dependent effect of dopamine on cell viability, morphology, and distribution, we chose two relevant cell types: primary cultures of PDLCs and the SH-SY5Y neuroblastoma cells line. Due to the neuroectodermal origin of the tooth, dental stem, and progenitor cells like PDLCs can differentiate into neuronal phenotype in vitro [31,54]. As SH-SY5Y cells can be differentiated into functionally mature neurons with dopaminergic characteristics, this cell line is a suitable model for dopaminergic neurons [37,39,55]. In addition, the presence of different types of dopamine receptors was detected on both dental stem cells [26], and SH-SY5Y cells [55]; therefore, dopamine can exert a direct effect on both PDLCs and SH-SY5Y cells.

Figure 2 shows the changes in viability of PDLCs (Figure 2a) and SH-SY5Y cells (Figure 2b) on the four different hydrogel types during the 2-week-long experiment. The data measured on day 1 depend mainly on cell adhesion while those of later time points refer to cell proliferation. Therefore, the viability data on day 1 on the hydrogel without dopamine were chosen as reference values.

**Figure 2 gels-08-00065-f002:**
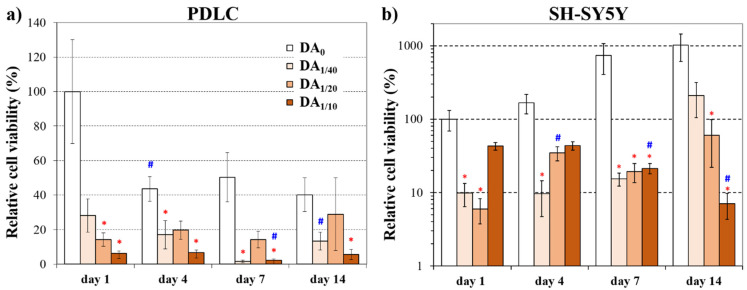
Relative cell viability of PDLCs (**a**) and SH-SY5Y cells (**b**) on the hydrogels 1, 4, 7, and 14 days after cell seeding. The viability was normalized to the value measured on the dopamine-free (DA_0_) gel on day 1. In case of the SH-SY5Y cells, a logarithmic scale was applied, as this cell line originates from a tumor and it grows much faster than the PDLC cells. * significant difference (*p* < 0.05) compared to the DA_0_ gel at the same time point. # significant difference (*p* < 0.05) compared to the same gel type at the previous time point.

We found that the highest dopamine concentration (DA_1/10_) in the hydrogel resulted in the lowest cell viability values of PDLCs (Figure 2a) compared to the other gels at each time point of the experiments. Although the viability was the highest on the DA-free gel (DA_0_) at day 1, remarkably lower viability data was measured on this gel type from day 4. This observation suggests that a great amount of PDLCs could adhere to the DA-free gels, but on this gel type, they were not able to proliferate. Previous studies on PASP hydrogels are also in line with our observations, and it has been described that these gels provide a suitable surface for cell adhesion and proliferation of PDLCs [15] and MG-63 osteosarcoma cells [8]. However, the viability of PDLCs did not change notably on DA-containing gels during the 14-day-long experiment. Therefore, the cells that adhered to these gels by day 1 remained viable for two weeks. Nevertheless, no significant difference was detected between the DA_0_ gel and DA_1/40_ or DA_1/20_ gel on day 14 while the viability on DA_1/10_ gel was significantly lower than those on DA_0_ gel at this time point. The lower cell viability on DA_1/10_ was probably caused by the high local DA concentration, which can exceed the lethal concentration of DA. For example, more than 100 µM concentration of dopamine induced cell death in the PC12 cell line, which consisted of a mixture of rat neuroblasts and eosinophilic cells [49]. Concentration-dependent cytotoxicity of dopamine on a rat mesencephalic/neuroblastoma hybrid cell line was also observed at concentrations between 5 and 50 µM [50]. The high DA concentration released from PASP-DA_1/10_ during cell experiments can explain the lower cell viability. Therefore, our results confirm the concentration-dependent, biphasic behavior of dopamine, as also described in the literature [29,49].

To date, several natural macromolecules like collagen [36], chitosan [4], and hyaluronic acid/alginate [35], as well as synthetic polymers like poly(glycolic acid), poly(lactic acid) [37], or poly(ethylene glycol) [38], have been tested as to whether they are suitable for cultivation of PDL cells. According to these results, PDLCs show good affinity to different natural and synthetic polymers and can adhere, proliferate, and even differentiate into neurogenic and osteogenic lineages on the previously mentioned matrices. In our previous publication, the proliferation and osteogenic differentiation of PDLCs on several types of PASP hydrogels without dopamine was also demonstrated. The viability of PDLCs shows a slight decrease on hyaluronic acid/alginate hydrogels scaffolds [35], which is in alignment with our viability results. The neurogenic differentiation ability of the PDLCs can also explain the vertical migration of the cells into the hydrogel which will be demonstrated later (in Section 2.4). Although several publications underline the role of dopamine in controlling cell behavior [21,22,23,24,25], as far as we know, this is the first study applying a dopamine-functionalized scaffold to affect PDLCs adhesion, migration, and proliferation.

In the case of the SH-SY5Y cells (Figure 2b), the viability on the DA-free gel was the highest compared to the other gel types during the whole experimental period. Besides, an enormous elevation was observed on this DA_0_ gel until day 14, suggesting a very intense proliferation. The viability data measured on the three DA-containing gel types were in a similar range at each time point until day 7. However, a remarkable difference was observed between them on day 14, when the highest value of DA_1/40_ gel reached the level of the DA_0_ gel. According to the literature, the SH-SY5Y neuroblastoma cell line can adhere and proliferate on various scaffolds consisting of natural polymers like alginate/gelatin [56], hyaluronic acid/poly(caprolactone) [57], collagen [41], gelatin [43], or nanocellulose [42]. In these studies, it was also demonstrated that the viability of the SH-SY5Y cells is highly dependent on the constitution of the scaffold material and also on the concentration of different molecules. To the best of our knowledge, dopamine-modified polymer-based materials have never been used for the cultivation of SH-SY5Y cells previously.

Comparing the viability values of the two cell types, we can conclude that for PDLCs, the DA_1/20_, while for the SH-SY5Y cell line, the DA_1/40_ gel proved to be the most suitable for long-term survival and proliferation. Although the 1/10 dopamine concentration seems to have a slightly promoting effect for adhesion of SH-SY5Y cells, this concentration is toxic for both cell types in the long term. At high concentrations, DA usually leads to a significant decrease in the number of viable cells regardless of cell type, while at low concentrations it can support cell attachment, proliferation, and differentiation, similarly to our results [22,58,59,60]. Since the same number of cells were seeded on the gel disk from both cell types, the explanation for the difference in the range of relative viability may be due to the tumor origin of the SH-SY5Y cell line. We observed that the tumor cell line proliferated at a higher rate than PDLCs even under standard culture conditions in Petri dishes before seeding them for these experiments. SH-SY5Y cells usually form clusters on three-dimensional hydrogel matrices [42,56,61], which can also be seen in Figure 3; this behavior can explain the discrepancy between the viability and microscopic results. Clusters can cause insufficient nutrition supply and hampered diffusion, therefore the WST-1 reagent cannot be converted to formazane by the mitochondria since it cannot reach the inner region of the clusters [42]. Consequently, the real number of viable cells can be better demonstrated by microscopic methods.

Using a phase-contrast microscope, cell growth on the surface of the gels can be visualized (Figure 3). A relatively higher number of PDLCs with healthy, fibroblast-like morphology was observed on the DA-free gel at days 7 and 14. This spindle-shaped cell form with a small number of long processes is typical for this cell type [30,31,32]. Regarding the DA-containing gels, only a few adhered cells could be seen on DA_1/10_ and DA_1/40_ gels during the whole experiment. However, the highest amount of fibroblast-like PDLCs was found on DA_1/20_ gels at each time point. The smaller cell size and the star cell shape or, at higher cell density, polygonal cell shape of the SH-SY5Y neuroblastoma cells are striking on the photomicrographs (Figure 3), which is in line with the description of the manufacturer (ECACC cell line). The highest amount of SH-SY5Y cells was found on DA_0_ gel at each time point. However, a large number of these cells could be detected under the phase-contrast microscope on day 14 on all DA-containing gel types as well with the lowest cell number on DA_1/10_ gels.

The aforementioned microscopic observations regarding cell morphology and proliferation on different gel types are in line with our viability results suggesting that while DA_1/20_ gels provide optimal DA concentration for adhesion and proliferation of both PDLCs and SH-SY5Y cells (Figure 2), higher DA-content can be toxic for both cell types.

These findings confirm the concentration-dependent effect of dopamine on cell viability, which is typical not only for dopamine treatment of cells cultured on the plastic surface, but also for cells grown on dopamine-containing hydrogel scaffolds. Biocompatibility studies with other sorts of dopamine-functionalized hydrogels are in accordance with our results, as none of these scaffolds have proved to be toxic. Extracts of dopamine-modified poly(ethylene glycol) hydrogels were demonstrated to have no toxic effect on mouse fibroblasts [60]. However, in that study, the DA content of the gels used for cell experiments was fixed and only the influence of the sterilization method on cell viability was investigated. In line with our results, the viability of human primary fibroblasts and embryonic fibroblast cell lines was slightly reduced after treatment with dopamine-modified alginate [25]. During development of a novel bioadhesive, gelatin-dopamine conjugates did not prove to be cytotoxic for human dermal fibroblasts [62]. Similarly to us, they observed that the adhesive property could be improved by increasing DA content only until a maximum point, with higher DA content leading to a decrease in adhesiveness. Finally, solutions with different concentrations of dopamine-modified poly(α,β-aspartic acid) derivatives did not influence the viability of mouse embryonal fibroblast cells [63]. In this work, in contrast to our study, only the cytotoxicity of the DA-modified polymer solutions with different concentrations was investigated and not of the prepared hydrogels.

The difference in the amount of covalently bound DA results in different free DA concentration and also different DA release kinetics. Both bound and free DA affect the microenvironment of the cells and induces different cell behavior by different concentration gradient of DA in the hydrogels. Nanoscale forces play a key role in tissue engineering [64], as can be seen in the cell morphology and viability, but also in the cell migration results.

### 2.4. Three-Dimensional Distribution of PDLCs and SH-SY5Y Cells on PASP-DA Hydrogels

The effect of the polymer solutions on cell viability, proliferation, and morphology is different than the effect of the hydrogels since the cells can uptake the polymers in solutions. However, several results show that the cells growing on three-dimensional polymer matrices can digest the bonds in the polymer matrix which may lead to the uptake of polymers or small molecules and induce vertical cell migration [65,66,67]. Since phase-contrast microscopy shows mainly the cell morphology on the hydrogel surfaces, but cannot provide information about the three-dimensional distribution of the cells and three-dimensional structure of the hydrogels, two-photon microscopy was applied. To visualize the cells in the inner parts of the hydrogels, they were labeled with the Vybrant DiD vital dye before seeding (Figure 4).

As can be seen on the three-dimensional images (Figure 4c), the topography of the PASP-DA_1/20_ hydrogels significantly changed due to the cells (hydrogel structure before cell seeding can be seen in Appendix A) and several cavities were formed in every sample. These cavities are clearly observable both on the upper view (Figure 4a) and the cross-sectional view images (Figure 4b). The lack of the green signal indicates the presence of the cavities in the hydrogel matrices, while the red color shows that the cells can be found at another height than the hydrogel upper surface. This indicates that both PDLCs and SH-SY5Y cells were able to degrade and digest the hydrogel matrices facilitating their vertical migration into the hydrogel matrices. According to the two-photon microscopic analysis, no similar cavity formation was discovered on the PASP-DA_0_, PASP-DA_1/10_, and PASP-DA_1/40_ hydrogels at any measurement time (Appendix A). Therefore, not only are the aforementioned cellular viability, proliferation activity, and differentiation [49,50] DA concentration-dependent, but this phenomenon is also related to the migration properties of the cells toward the inner parts of the hydrogel matrix, observation of which has never been published before.

Although there are several examples for culturing SH-SY5Y [41,43,68,69,70] and PDLCs [15,36,71,72] on various hydrogels, the migration mechanisms of these cells have barely been described. In general, cells are able to migrate in 3D matrixes by different mechanisms, such as mesenchymal, amoeboid, and lobopodial movements [66]. The sort of migration mechanism is supposedly influenced by not just the cell type, but also by the environmental properties (topography of ECM, the molecular composition and concentration, stiffness, substrates, elasticity, rigidity, density, pore size, cellular interactions). The amoeboid cells do not align to ECM fibers by contractile forces while mesenchymal cells create special contacts with the ECM through their receptors. In addition, mesenchymal cells can proteolytically digest the ECM, and thereby create pathways for themselves during migration. A highly crosslinked matrix can induce lobopodial migration, which is a hybrid mechanism of the amoeboid and the mesenchymal migration [65,66]. According to these theories and our results, the vertical migratory mechanism of the PDLCs and SH-SY5Y cells into PASP-DA_1/20_ hydrogels could be mesenchymal or lobopodial. The observed cavities in the hydrogels indicate that cells may proteolytically digest the PASP polymers or the crosslinks in the matrix that can be induced by dopamine. In the case of the other hydrogel types without DA (Appendix A), no vertical migration was observed that shows the crucial role of the dopamine concentration in the vertical migration of the cells.

Nevertheless, migration facilitated by our DA-containing hydrogels could be in connection with the oxidative stress generated during metabolism of dopamine. It is well known that degradation of dopamine produces reactive oxygen species (ROS) [73], which can promote cell migration by redox regulation of key signaling pathways [74], or enhances migration of tumor cells, and thereby the invasion and metastasis of cancer [75]. Moreover, the important role of ROS signaling was demonstrated during neurodevelopment as a modulator of proliferation and migration as well [76]. It was described that the hexokinase enzyme located in mitochondrial membrane is not only a potent regulator of ROS signaling but also a novel target of dopamine signaling in neural progenitor cells [77]. Even in the case of SH-SY5Y human neuroblastoma cells, it was found that exposure to dopamine inhibited mitochondrial respiration [78]. Based on the abovementioned results, we suppose that PDLCs (with ectodermal origin and neuronal differentiation potential) and SH-SY5Y cells (consisting of immature neurons) may apply the same sort of mechanism for migration in the DA-containing PASP-hydrogels due to the release of DA and generation of ROS as the invading tumor cells. To quantify the vertical migration of the cells topographical analysis was carried out on the two-photon microscopic images. The extent of vertical cellular migration was calculated at each (x, y) point on the horizontal plane as the vertical distance between the hydrogel surface digested by the cells and an estimated original surface. The surface estimation was based on fitting a 3rd degree bivariate polynomial on the hydrogel surface and shifting its zero value to be equal to the mode of the hydrogel surface. This method is based on the assumption that cells (due to their relatively low number) affect only the shape of the surface locally and the large-scale features of the original hydrogel can be found by fitting a slowly varying function on the surface. The distance between the estimated original surface and the surface digested by the cells was further analyzed: negative peaks representing local maxima of cellular migration distance (locally deepest points of the surface) were found and the peaks’ topographic prominences were also calculated. Topographic prominence represents how much a peak’s depth stands out compared to its surroundings: it is the difference in distance between a peak and (in the case of negative peaks) its highest contour. Intuitively, a gel surface with deep peaks but low prominences might correspond to a large group of cells migrating together creating a large cavity, while the same depth values with high prominences would represent cells that at some point migrate in smaller groups or individually. The prominence and depth data are illustrated in Figure 5.

According to these results, most of the cavities are in connection with each other and only a few of them are separated. This may confirm that the cells, especially the SH-SY5Y cells, are organizing themselves into clusters at first with high cell density [61]. These clusters are clearly observable on the phase-contrast microscopic images (Figure 3, SH-SY5Y day 14). This cluster formation inhibits intracellular communication and vertical migration of the SH-SY5Y cells, which was also described by Q. Chen and co-workers [56]. The three-dimensional images (Figure 4c) also prove this statement, where the cavities made by the cells are also connected to each other, and they form extended valleys in the hydrogel matrices. The distribution of the absolute depth of the pixels follows a Gaussian distribution (Figure 5b) in the case of every sample. To compare the different cell types and measurement days with each other, the average depth of the peaks was calculated (Table 3).

According to our results (Table 3 and Figure 5b), the PDLCs were able to migrate deeper into the polymer gel matrix than the SH-SY5Y cells. The vertical migration ability of the PDLCs is time-dependent, since the cells were able to reach almost double deepness after 14 days of seeding than after 7 days. The average size of a PDLC is around 30 μm [32]; therefore, the deepest cavities in the hydrogel after 7 days correspond to approximately 2–3 cell layers, while after 14 days they reached a depth of around 4–5 cell layers. The different migration data of PDLCs and SH-SY5Y cells can be explained by the significant size difference between the two cell types (the average size of the SH-SY5Y is around 12 μm) [40].

Although the mechanism behind the migration of the SH-SY5Y cells has not been clarified yet, similar behavior has been described in several studies. G. Akcay and co-workers investigated the vertical migration of SH-SY5Y cells into photopolymerized poly(ethylene glycol)-diacrylate hydrogels [61]. According to their results, the migration of the cells was induced by the stiff-to-soft material transition, and the cells were able to migrate into the hydrogels up to 60 μm. They concluded that SH-SY5Y cells in the 3D hydrogels exhibit a spherical morphology—unlike in 2D, where they have a flat shuttle-like shape with long pseudopods—due to the mechanical support and the limited space to spread [56,61]. Similar behavior of PDLCs has never been described in any type of hydrogel matrix.

The vertical migration of the cells is a crucial behavior for tissue replacement, and makes these hydrogels a very promising candidate for this application. The novel method we developed for the representation, quantification and analysis of the vertical migration can be used to describe this kind of changes in other sorts of matrices in the future. Currently, there is no similar method available in the literature.

Our future plan is to further optimize the composition of the DA-containing PASP gels in order to make them more suitable for proliferation and migration of cells. As the use of antioxidants could prevent dopamine-induced cell death in many cases [79], one of the promising ways would be the incorporation of antioxidants into these gels.

## 3. Conclusions

In this work, poly(aspartic acid) hydrogels with different dopamine content (PASP-DA_1/40_, PASP-DA_1/20_, PASP-DA_1/10_) were synthesized for cell cultivation. The chemical constitution of the hydrogels was confirmed and the exact dopamine content after hydrogel preparation was determined. To test the hydrogels as scaffolds, PDLCs and SH-SY5Y cell lines were used. Both cell types showed a concentration-dependent reaction to the presence of dopamine in the scaffold: dopamine at low concentration (PASP-DA_1/40_ or PASP-DA_1/20_) did not affect cell viability but at higher concentration (PASP-DA_1/10_) it induced apoptosis and lowered the number of viable cells. We observed for the first time in the literature the unique migration phenomena of the cells into the hydrogels induced by a specific dopamine concentration (PASP-DA_1/20_) in the case of both cell types. Either PDLCs or SH-SY5Y cells were able to degrade the hydrogel matrices, making the way free for themselves to migrate into the hydrogels.

To quantify the depth of the cell migration, a new method was developed based on the two-photon microscopic measurements of the hydrogel matrices. According to these results, PDLCs were able to penetrate into the hydrogel matrix even to a depth of 150 µm, which is equal to the size of 4–5 cell layers, while SH-SY5Y showed lower migration capacity that can be explained by the neurogenic origin of the cells.

Consequently, based on our results and due to the wide differentiation capacity of PDLSCs and other types of progenitor cells, several sorts of artificial tissues can be constructed in the future. Our new image analysis method can be used widely to quantify the three-dimensional migration of the cells in various matrices.

## 4. Materials and Methods

### 4.1. Preparation of Dopamine Modified Poly(succinimide)s

At first, poly(succinimide) (PSI) was synthesized as described in our previous publications [8,12,13,14,15,80]. In the second step, dopamine (DA) modified poly(succinimide) (PSI-DA) was synthesized with different dopamine (Sigma Aldrich; 99.9%) content (Figure 6, Step 1). The detailed method of synthesis was published previously [8,13,15,18] and can also be found in the Appendix A, together with the exact constitution of the reaction mixtures (Appendix A).

### 4.2. Chemical Characterization by Nuclear Magnetic Resonance (NMR) Spectroscopy

For ^1^H-NMR characterization, the PSI-DA polymers were treated with the same materials, i.e., pH = 8 imidazole buffer (imidazole: ACS reagent, ≥99%, Sigma-Aldrich, Budapest, Hungary), citric-acid*H_2_O (ACS reagent, ≥99.9%, VWR), sodium chloride (99–100%, Sigma-Aldrich) that were used for hydrolyzation of PSI-based gels into PASP gels. All ^1^H NMR spectra were obtained using a JEOL SC400 spectrometer (JEOL Ltd., Riken, Tokyo, Japan) operating at 400 MHz. Then, 25 mg of polymer powders were dissolved in 0.6 mL of D_2_O in 5 mm NMR tubes. All spectra were recorded at 23.5 °C ± 0.5 °C and tetramethylsilane was used as an internal standard. The pulse angle was 45°, and a 2 s delay was applied with 8 kHz spectral width, while 16 K data points and 16 scans were done at each measurement. To evaluate spectra, the ACD/Labs NMR software was used.

### 4.3. Preparation and Characterization of DA Containing PASP-Based Hydrogels

To prepare DA-containing hydrogels, PSI-DA polymers were crosslinked with 1,4-diaminobutane (DAB) and cystamine (CYS) according to our previously published method [8,12,15], where pure PSI was used instead of PSI-DA polymers. The reaction steps can be seen in Figure 6. To form thiolated PASP hydrogels, PSI-DA gels were treated with pH = 8 imidazole buffer, and afterwards with pH = 8 0.1M dithiothreitol (DTT) solution (≥99%, Sigma). For the cell experiments, the gel samples were treated with phosphate-buffered saline (PBS) (pH = 7.5, c = 150 mM) (Sigma Aldrich). Since the PSI was modified with different amounts of DA (statistically every 10th, 20th, and 40th monomer was modified with dopamine), the dopamine content of the hydrogels was indicated in their abbreviation PASP-DA_X_, where X shows the molar ratio of the modified monomer units compared to the whole number of monomers. The control hydrogels (DA_0_) were not modified with DA, but they contained the same amount of crosslinkers and thiol side-chains as the DA-modified hydrogels. The details can be seen in the Appendix A.

To determine the pore size of the hydrogels, the samples were freeze-dried and investigated by scanning electron microscopy (SEM, ZEISS, Budaörs, Hungary). Microscopic images were taken using a ZEISS EVO 40 XVP scanning electron microscope equipped with an Oxford INCA X-ray spectrometer (EDS, Oxford Instruments, Abingdon, UK). The applied accelerating voltage was 20 kV. The samples were fixed on a special conductive sticker. Before the examination, the samples were sputter-coated with palladium in a thickness of 20–30 nm with a 2SPI Sputter Coating System. The average pore size of the samples was determined by the Fiji (ImageJ v1.53) image analysis program from 20 parallel measurements.

The chemical structure of the hydrogels was investigated by Fourier transform infrared spectroscopy (FTIR, JASCO, Budapest, Hungary) according to the method described in our previous paper [14].

### 4.4. Determination of the DA Content of the PASP Hydrogels

To determine the released amount of dopamine, rectangular gel films were immersed into pH = 8 imidazole-based buffer and after 1 day the UV absorbance was measured by an Agilent 8453 spectrophotometer (Agilent, Budapest, Hungary) at 280 nm. Following the measurement, the buffer was changed, and 24 h later the absorbance was measured again. The amount of the released dopamine was determined by using a calibration curve (Appendix A).

### 4.5. Culturing of PDLCs and SH-SY5Y Cells

PDLCs were obtained from impacted human wisdom teeth removed at the Department of Dentoalevolar Surgery of the Semmelweis University (Budapest, Hungary) under ethical guidelines approved by the Ethical Committee of the Hungarian Medical Research Council. This study was approved by the Semmelweis University Regional and Institutional Committee of Science and Research Ethics, the number of the ethical permissions are 17458/2012/EKU and 25459/2019/EKU. Isolation of PDLCs was performed according to our previously published protocol [15,81].

The PDLCs were cultured in Minimal Essential Medium Alpha (αMEM, Gibco, Thermo Fisher, Waltham, USA) supplemented with 10% fetal bovine serum (FBS, Gibco), 2 mM L-glutamine (Gibco), 100 units/mL penicillin and 100 mg/mL streptomycin (Gibco).

The SH-SY5Y human neuroblastoma cell line was purchased from Sigma-Aldrich (Budapest, Hungary) and maintained according to the manufacturer’s protocol. The growth medium of these cells consisted of a 1:1 mixture of Eagle’s MEM and Ham’s F12 medium supplemented with 15% FBS (Gibco), 2 mM L-glutamine (Gibco), 100 units/mL penicillin, and 100 mg/mL streptomycin (Gibco), as well as 1% non-essential amino acids (NEAA, Gibco).

### 4.6. Cell Viability Assay

Before the cell seeding, hydrogel discs were incubated in the cell culture medium for 2 h with a medium change after 1 h. Subsequently, the gel discs were placed into the appropriate cell culture plates and sterilized by UV irradiation for 1 h.

For measuring cell viability, gel discs with a diameter of 6 mm were placed into Nunclon Sphera 96-well microplates with surface treatment allowing very low cell attachment (Nunc, Roskilde, Denmark). Then, 20,000 PDLCs in 200 μL culture medium were seeded on each gel disc. In case of the SH-SY5Y cell line, 10,000 cells were seeded in 200 μL culture medium on each gel disc since we observed that this cell type does not tolerate high cell density. With each gel type, 5 parallel experiments were carried out and the whole experiment was repeated 3 times with new cell cultures of both cell types. Cell viability was assessed after 1-, 4-, 7-, and 14-day-long cultivation on the gel discs using the cell proliferation reagent WST-1 (Roche, Basel, Switzerland) as we described recently [8].

### 4.7. Cell Morphological Studies

For the investigation of cellular morphology, 40,000 PDLCs or 20,000 SH-SY5Y cells in 800 μL culture medium were seeded on gel discs with a diameter of 10 mm placed into 24-well plates (Sigma-Aldrich, St. Louis, MO, USA). After 1, 4, 7, and 14 days, the cell cultures were observed under a phase-contrast microscope (Nikon Eclipse TS100, Nikon, Tokyo Japan) with a 10× objective. Photomicrographs were taken with a high-performance CCD camera (COHU, USA) applying the Scion image software.

To visualize the 3-dimensional distribution of the cells in the hydrogels, Vybrant DiD (Molecular Probes, Thermo Fisher, Waltham, MA, USA) labeling and 2 photon microscopy were applied according to our previous paper [8].

### 4.8. Analysis of the Two-Photon Microscopic Images

To determine the penetration depth of the cells in the hydrogel matrices, image analysis was carried out using the Python (version 3.7.6) Programming Language. First, color deconvolution was used on the microscopic image stacks to reduce the effects of fluorescence crosstalk and create an image stack containing only the hydrogel without the cells. Small bright imaging artifacts were removed using the outlier removal algorithm of NIH ImageJ implemented in python. The surface of the gel was estimated by finding the depth value (z distance) corresponding to the half-maximum intensity point for all pixels in the (x, y) plane. A third-degree polynomial was fitted to the resulting surface depth map to estimate the shape of the original hydrogel surface before the cells were applied. The difference between the depth map and this estimated original surface corresponds to the effect of the cellular penetration on the shape of the hydrogel surface. This penetration image was calculated and after Gaussian smoothing and subtracting its mode (which acts as an estimate for zero depth) was used for further processing and analysis as follows. Negative peaks with a depth of at least 5 µm were found using the h_minima function in the scikit-image python package [82]. Depth values and peak prominences were also calculated for each peak. Areas near the edges of the images with very low signal-to-noise ratios were excluded using a binary mask. Histograms of peak depths, prominences, and penetration image values were constructed. 3D Slicer 4.10.2 [83] was used to create 3D rendered images from the image stacks to further visualize the hydrogel surface along with the cells.

### 4.9. Statistical Analysis

Statistical evaluation of the viability data was carried out applying the STATISTICA 10 software using Kruskal-Wallis non-parametric ANOVA followed by median test. In the case of *p* < 0.05, a difference was considered as statistically significant.

## Figures and Tables

**Figure 1 gels-08-00065-f001:**
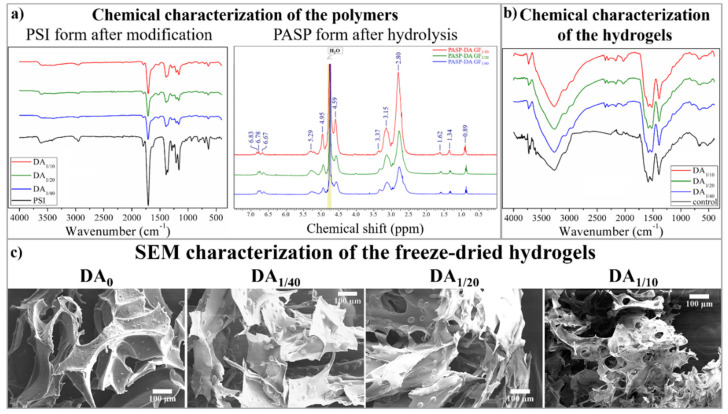
FTIR and NMR spectra of PSI and PASP-based polymers (**a**) and FTIR spectra of PASP-based hydrogels (**b**). SEM images of the freeze-dried dopamine-containing and dopamine-free control (DA_0_) gel (**c**).

**Figure 3 gels-08-00065-f003:**
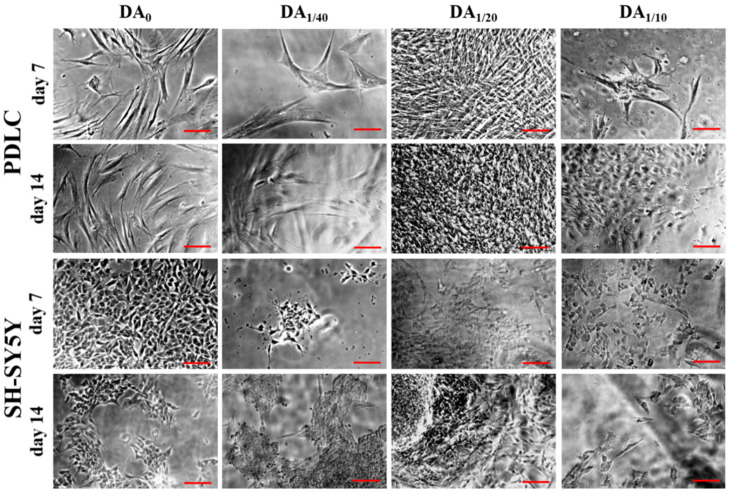
Phase-contrast microscopic images of PDLCs and SH-SY5Y cells growing on the 4 different gel types for 1 or 2 weeks. The scale bars indicate 100 µm.

**Figure 4 gels-08-00065-f004:**
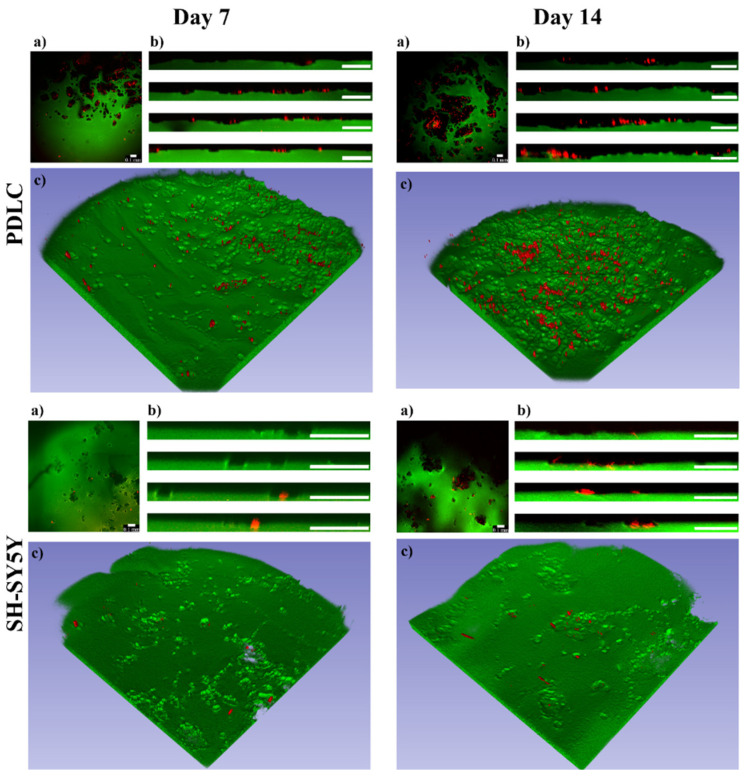
PDLC and SH-SY5Y cell (red) distribution on the PASP-DA_1/20_ hydrogels (green) on days 7 and 14. The upper view (**a**), cross-sectional view (**b**), and three-dimensional images (**c**) were constructed from Z-stack measurements of the hydrogels. Scale bars indicate 0.1 mm.

**Figure 5 gels-08-00065-f005:**
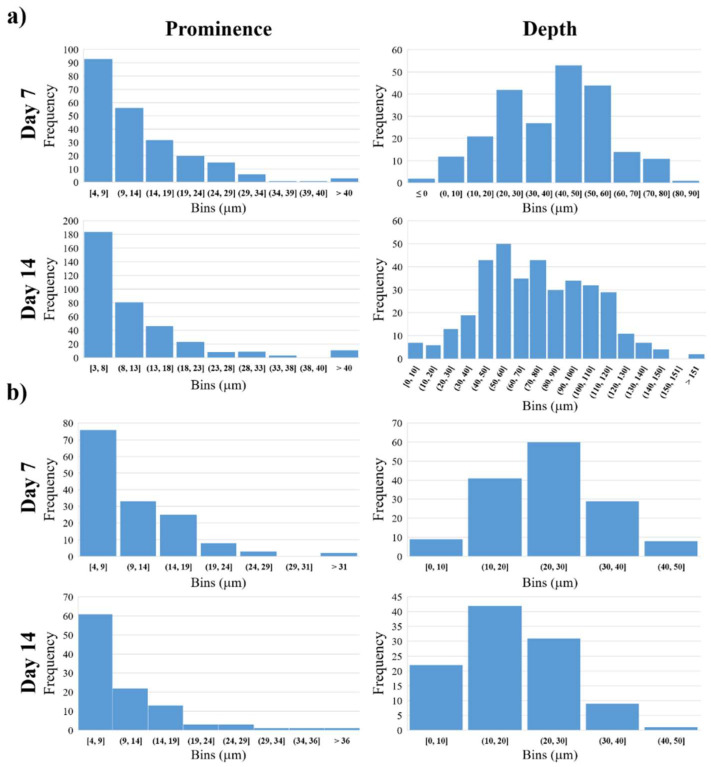
Distribution of the prominence and depth of the migration distance from the fitted surface in the PASP-DA_1/20_ in the case of PDLC (**a**) and SH-SY5Y (**b**).

**Figure 6 gels-08-00065-f006:**
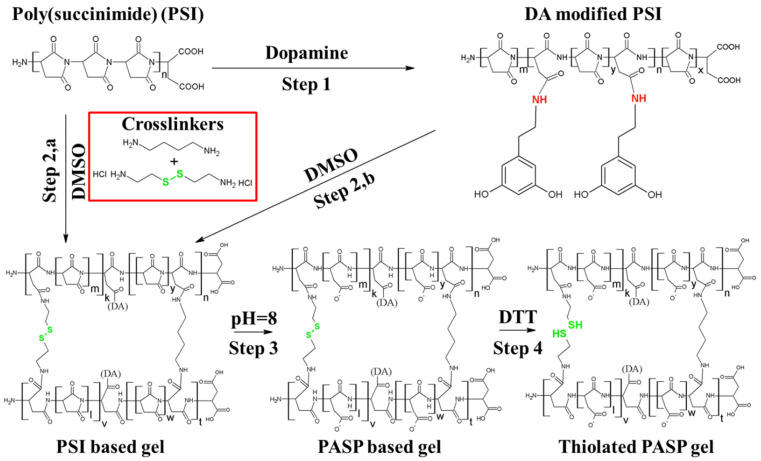
Chemical synthesis of dopamine-containing gels.

**Table 1 gels-08-00065-t001:** Average pore size of the hydrogels with different dopamine content.

	DA_0_	DA_1/40_	DA_1/20_	DA_1/10_
Average pore size	120 ± 13 μm	121 ± 25 μm	110 ± 35 μm	87 ± 27 μm

**Table 2 gels-08-00065-t002:** The released amount of DA and DA concentration of the hydrogels at the time point of cell seeding, and the mass swelling degree of the hydrogels.

Sample	Released Amount of DA (%)	DA Concentration in the Hydrogels after Hydrolysis (mmol/L)	Mass Swelling Degree
PASP-DA_1/40_	48.7 ± 4.81	4.73 ± 0.31	29.3 ± 0.6
PASP-DA_1/20_	58.3 ± 3.43	9.43 ± 0.78	21.8 ± 0.4
PASP-DA_1/10_	54.2 ± 3.44	23.96 ± 2.8	18.3 ± 0.3

**Table 3 gels-08-00065-t003:** Average depth of the migration distance from the fitted surface in μm on PASP-DA_1/20_ in the case of different cell lines.

	Average Depth (μm)	
	PDLC	SH-SY5Y
Day 7	39 ± 18	24 ± 9
Day 14	75 ± 30	18 ± 8

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
