# Peer review of "Analysis of Three-Dimensional Cell Migration in Dopamine-Modified Poly(aspartic acid)-Based Hydrogels"

_gels, 2022, doi:10.3390/gels8020065_

Round 1

Reviewer 1 Report

This study investigate three-dimensional cell migration in dopamine modified poly(aspartic acid) based hydrogels
1. Why the authors did not discuss the role of  stress oxidative cell migration 
2.It is suggested to discuss the role of mitochondria
3. Please add your future perspective and suggestion about the possible role of antioxidant therapy.
4.Please add combination of nanomedicine with stem cell and also different type of hydrogels in tissue repair 
5. It is suggested to use the following paper for the discussion part :
-Ahmadian, Elham, et al. "Hyaluronic acid hydrogel nanoscaffolds: Production and assessment of the physicochemical properties." Eurasian Chemical Communications 2.1 (2020): 51-58.

Reviewer 2 Report

The author observed firstly in the litera-552 ture the unique migration phenomena of the cells into the hydrogels induced by a specific 553 dopamine concentration (PASP-DA1/20), in the case of both cell types. In order to verify the validity of current results, authors need to compare their results with reported experimental data in literatures.  However, if the authors are willing to make the revisions according to my comments, I would be glad to re-review this manuscript. Here are my detailed comments:

  1. Why do you use the PDLCs or SH-554 SY5Y cells ?
  2. In order to verify the obtained results, a comparison with the literature is necessary using the table form.
  3. Figure 4 needs the scale for each picture.
  4. Please, expand the conclusions in relation to the specific goals and the future work.

Reviewer 3 Report

I find the article interesting and after a review of the introduction I would be happy to review it again. 
